# Study on the Low-Damage Material Removal Mechanism of Silicon Carbide Ceramics Under Longitudinal–Torsional Ultrasonic Grinding Conditions

**DOI:** 10.3390/mi16091048

**Published:** 2025-09-13

**Authors:** Junli Liu, Zhenqi Ma, Yanyan Yan, Dengke Yuan, Yifan Wang

**Affiliations:** School of Mechanical and Power Engineering, Henan Polytechnic University, Jiaozuo 454000, China; 17639509225@163.com (Z.M.); 15617226992@163.com (D.Y.); 15038261552@163.com (Y.W.)

**Keywords:** SiC ceramics, ultrasonic vibration-assisted grinding, surface quality, the brittle–plastic transition, critical cutting depth

## Abstract

In order to achieve the high-performance machining of silicon carbide (SiC) ceramics, longitudinal–torsional ultrasonic vibration (LTUV) was introduced into precision machining, and a systematic investigation into the effects of various process parameters on the critical cutting depth and surface quality was conducted. This investigation was undertaken with a view to exploring the ultrasonic vibration-assisted grinding mechanism of SiC ceramics. Firstly, the kinematic model of single abrasive grain trajectory and the maximum unaltered cutting thickness during longitudinal–torsional ultrasonic vibration-assisted grinding (LTUVG) was established to explore its unique grinding characteristics. On this basis, the theoretical modeling of critical cutting depth in SiC ceramics under LTUVG conditions was developed. This was then verified through longitudinal–torsional ultrasonic scratching (LTUS) experiments, and the theoretical analysis and test results prove that compared with normal scratching, the quality of SiC grooves are significantly improved by means of LTUS. During LTUS experiments, the dynamic fracture toughness, strain rate of SiC, and high-frequency ultrasonic excitation significantly enhances SiC performance, increasing the critical cutting depth and expanding the plastic removal region, so it is easy for LTUVG to yield the better surface quality in machined SiC ceramics, which provides important scholarly support for achieving the low-damage machining of SiC ceramics.

## 1. Introduction

Silicon carbide (SiC) ceramics exhibit outstanding properties including high hardness, high thermal conductivity, high strength, low density, good electrical conductivity, excellent chemical resistance, and superior wear resistance. These advantages make them widely applicable in fields such as aerospace, mechanical engineering, and semiconductors [1,2]. Currently, grinding is the primary machining method for the precision machining of SiC ceramics. However, due to the fact that SiC ceramics are categorized as brittle and hard materials, it is difficult to efficiently obtain high-integrity manufactured products. In particular, during the traditional grinding, various types of defects and damages are easily generated, such as deterioration layers, subsurface cracks, and micro-cracks on the surface, which not only decrease quality and shape precision but also decrease fatigue strength, greatly shortening product service life [3,4].

In order to achieve the low-damage machining of SiC ceramics, researchers have applied ultrasonic vibration machining to the common grinding. Lu et al. [5] examined the response of critical depth and grinding force in SiC to both ultrasonic and normal scratching, and the results indicated that ultrasonic scratching produced a significant reduction in grinding force and improved the critical depth compared with normal scratching. Qiao et al. [6] examined the response of silicon nitride ceramics to vibration-assisted and normal scratching, with focus on removal mode transitions and crack suppression effectiveness, and the test results showed that crack suppression between scratches diminishes with increasing distance between scratches. An increase in the critical brittle–plastic transition load is achieved through vibration-assisted scratching, while ultrasonic vibration can improve crack suppression at the same distance between scratches compared with normal scratching. Cao et al. [7] investigated the influence of conventional and longitudinal ultrasonic scratching on scratching force and the critical cutting depth, and the test results revealed that the scratching force under ultrasonic scratching was lower than that in conventional scratching, and the critical depth of SiC ceramics was increased by approximately 56.25% compared with normal scratching. Li et al. [8] investigated the effect of ultrasonic scratching and normal scratching on the critical depth of SiC ceramics under various cutting depths, and the results indicated that the critical depth of SiC ceramics is increased and the subsurface crack depth is significantly reduced after ultrasonic scratching compared with normal scratching. Zhang et al. [9] investigated the material removal mechanism of elliptical ultrasonic vibration-assisted grinding (EUVG) and performed a comparative test between ultrasonic scratching and normal scratching (NS); the results showed that EUVG significantly reduces the grinding force and specific energy and increases the material removal rate of SiC. Moreover, the specific energy of grinding decreases with the increase in thickness of the unaltered chip and increases with grinding wheel speed compared with CG. In general, scholars are mostly focused on the comparisons between different machining methods, and there is limited research on the brittle–plastic transition of SiC ceramics from the perspective of the material strain rate under ultrasonic vibrations.

To advance the ultra-precision machining of high-performance SiC ceramic components, LTUV was implemented, and the critical depth model of SiC ceramics under ultrasonic vibration conditions was established by taking the material strain rate efficiency into account. In addition, the scratching force on SiC ceramics and the resulting surface damage were studied, which may provide a theoretical basis and technological support for the low-damage machining of the brittle–hard material.

## 2. Modeling the Critical Cutting Depth of SiC Under LTUVG Conditions

The LTUVG system is shown in Figure 1. As shown in Figure 1, in LTUVG, the workpiece has a feed motion at a speed of $v_{w}$ along the X-axis of the diamond grinding wheel, and the ultrasonic vibrations are axially imparted to the diamond grinding wheel and the circumferential directions, while the diamond grinding wheel rotates around the spindle at speed $v_{s}$.

According to Figure 1, assuming the abrasive grain cuts into the workpiece at point $m_{1}$, it cuts out the workpiece at point $m_{2}$ when the grinding wheel rotates through angle θ, and the motion trajectory can be shown as follows:(1)$$\left\{\begin{array}{l} x\left(t\right)=R\sin\theta +v_{w}t\\ y\left(t\right)=R\cos\theta \\ z\left(t\right)=A_{b}\sin\left(2\pi f_{b}t\right) \end{array}\right.$$
where R is radius of the grinding wheel, mm; $v_{w}$ is feed rate, mm/s; t is machining time, s; $A_{b}$ is longitudinal ultrasonic amplitude, μm; and $f_{b}$ is longitudinal ultrasonic frequency, Hz.

According to Figure 1, $v_{s}$ can be shown as follows [10,11]:(2)$$\nu_{s}=\nu_{a}+\nu_{b}=\omega R+2\pi f_{a}A_{a}\cos(2\pi f_{a}+\varphi )$$
where $v_{a}$ is rotational speed of the grinding wheel, mm/s; $v_{b}$ is caused by torsional ultrasonic vibration, μm/s; ω is angular velocity of the grinding wheel, rad/s; $A_{a}$ is amplitude in the torsional direction, μm; $f_{a}$ is torsional frequency, Hz; and φ is phase difference between axial and circumferential vibrations.

Figure 2 shows the schematic representation of the maximum unaltered cutting thickness ($a_{g}$) of a single abrasive particle during LTUVG. As shown in Figure 2, the cross-sectional area of the chip is correlated with the cutting arc length $l_{g}$ of the abrasive grains and $a_{g}$. It can be represented as follows:(3)$$E\left(A_{g}\right)=E\left(a_{g}\cdot l_{g}\right)$$
where $A_{g}$ is cross-sectional area of chip, ${\mu m}^{2}$.

MMR can be shown as follows [12]:(4)$$MMR=\sqrt{N_{d}}\nu_{s}E\left(a_{g}\right)l_{g}$$
where $N_{d}$ is effective quantity of abrasive particles.

The effective quantity of abrasive particles $N_{d}$ per unit grinding area can be obtained [13]:(5)$$N_{d}={\left(\frac{6\nu_{g}}{\pi d_{g}^{3}}\right)}^{2/3}$$
where $v_{g}$ represents the volume proportion of diamond abrasive particles within the grinding wheel, under the conditions of a 100% diamond concentration, $v_{g}=25\%$, and $d_{g}$ is the abrasive particle diameter.

According to Equations (4) and (5), the material removal rate of SiC during LTUVG can be shown as follows:(6)$$MMR={\left(\frac{6\nu_{g}}{\pi d_{g}^{3}}\right)}^{1/3}\nu_{s}E\left(a_{g}\right)l_{g}$$

During the grinding of single abrasive particle, MMR can be shown as follows:(7)$$MMR=a_{p}\nu_{w}b$$
where b represents the grinding width, mm.

According to Equations (6) and (7), $a_{g}$ can be shown as follows:(8)$$a_{g}=\frac{\pi^{1/3}a_{p}\nu_{w}d_{g}}{{\left(6\nu_{g}\right)}^{1/3}\nu_{s}l_{g}}$$

Substituting Equation (2) into Equation (8), the maximum unaltered cutting thickness of SiC during LTUVG can be shown as follows:(9)$$a_{g}=\frac{\pi^{1/3}a_{p}\nu_{w}d_{g}}{{\left(6\nu_{g}\right)}^{1/3}\left[\omega R+2\pi fA_{a}\cos(2\pi f_{a}t+\varphi )\right]l_{g}}$$

According to Equation (9), the maximum unaltered cutting thickness of SiC under the LTUVG is related to grinding parameters, cutting arc length, and ultrasonic amplitude; it decreases with the increase in grinding speed, ultrasonic amplitude, and cutting arc length and increases with the grinding depth and feed rate.

The maximum unaltered cutting thickness of single abrasive particle is associated with grinding velocity and the strain rate of SiC [14], and it can be shown as follows [15]:(10)$$\dot{\varepsilon }=\frac{v_{s}}{a_{g}}$$

Substituting Equation (9) into Equation (10), $\dot{\varepsilon }$ can be shown as follows:(11)$$\dot{\varepsilon }=\frac{\nu_{s}{\left(6\nu_{g}\right)}^{1/3}\left[\omega R+2\pi fA_{a}\cos(2\pi f_{a}t+\varphi )\right]l_{g}}{\pi^{1/3}a_{p}\nu_{w}d_{g}}$$

The relationship between the static fracture toughness $K_{IC}$ and the dynamic fracture toughness $K_{ID}$ of SiC under high strain rates can be shown as follows:(12)$$K_{ID}=\left(m+n\ln\dot{\varepsilon }\right)K_{IC}$$
where m and n are material constants and these can be shown as −1.64 and 0.675, respectively [16].

According to Equations (11) and (12), $K_{ID}$ can be shown as follows:(13)$$K_{ID}=K_{IC}\left[-1.64+0.675\ln\left(\frac{\nu_{s}{\left(6\nu_{g}\right)}^{1/3}\left[\omega R+2\pi fA_{a}\cos(2\pi f_{a}t+\varphi )\right]l_{g}}{\pi^{1/3}a_{p}\nu_{w}d_{g}}\right)\right]$$

The $K_{ID}$ is intricately associated with the critical depth. According to reference [17], the critical depth of hard–brittle materials can be shown as follows:(14)$$d_{c}=\left(\frac{E}{H}\right){\left(\frac{K_{ID}}{H}\right)}^{2}\frac{\left(1-{\nu_{g}}^{2}\right)}{d_{g}}$$

Substituting Equation (13) into Equation (14), the critical depth of SiC ceramics under LTUVG can be shown as follows:(15)$$d_{c}=K_{IC}^{2}\left(\frac{K_{ID}}{H^{3}}\right)\frac{\left(1-\nu_{g}^{2}\right)}{d_{g}}{\left[-1.64+0.675\ln\left(\frac{\nu_{s}{\left(6\nu_{g}\right)}^{1/3}\left[\omega R+2\pi fA_{a}\cos(2\pi f_{a}t+\varphi )\right]l_{g}}{\pi^{1/3}a_{p}\nu_{w}d_{g}}\right)\right]}^{2}$$

According to Equation (15), the critical depth of SiC under LTUVG is related to grinding parameters, ultrasonic amplitude, cutting arc length and strain rate, and the critical depth of SiC reduce due to the introduction of LTUV; meanwhile, the critical depth increases with grinding speed, ultrasonic amplitude, cutting arc length, and strain rate and decreases as the grinding depth and feed rate increase.

## 3. LTUS Test

To further validate the theoretical model, LTUS tests were conducted on SiC, examining the influence of velocity and amplitude on the force applied during the scratching process under the variable cutting depths. To validate the influence of various parameters on critical depth, the surface morphologies of SiC ceramics produced by ultrasonic scratching were compared with those from normal scratching using a Super-Depth Electron Microscope (VHX2000, Keyence Corp., Osaka, Japan).

### 3.1. Construction of Test Platforms

The scratching platform in the LTUV system is shown in Figure 3. As can be seen in Figure 3, the platform consists of a three-axis vertical machining center (VMC850E) and the LTUV system; the LTUV system consists of ultrasonic generator (35 kHz), wireless transmission system, piezoelectric transducer, horn with longitudinal torsional vibration, and BT40 toolholder. The ratio of longitudinal torsional amplitude of the horn is 1:1. During the test, ultrasonic scratching was performed when the switch of the ultrasonic generator was turned on, and normal scratching was performed when it was turned off.

The main performance parameters of the workpiece are shown in Table 1 Before the tests, the surface of SiC ceramic to be scratched was polished, and its surface roughness, $R_{a}$, after polishing was less than 1 nm. The radius of diamond indenter was 1 μm, the angle of the indenter relative to the edges was 120°, and the angle of the indenter cone was 120°. The workpieces used in the experiment is shown in Figure 4.

### 3.2. Experimental Design

The workpiece was installed on the wedge platform, and the angle between the platform and the horizontal workbench of the machine tool was set to 0.07°. The scratch experiment with variable cutting depth was conducted, where the scratch depth continuously increased as the diamond pen moved at a feed rate of 1 μm/s; the ultrasonic scratching is shown in Figure 5.

During the scratching, both the tangential force and normal force were recorded in real time by means of a dynamometer (Kistle 9257B Kistler Instrumente AG, Winterthur, Switzerland), and the sampling rate of the dynamometer was 80 kHz. The parameters of the scratching test, assuming that the frequency and amplitude of the horn remain stable during the scratching process, are shown in Table 2.

After the test, the machined workpiece was cleaned in an ultrasonic cleaner with acetone solution for 15 min, and then the morphology of the machined surface was observed using a Super-Depth Electron Microscope (VHX2000 Keyence Corp., Osaka, Japan).

### 3.3. Experimental Results and Analysis

The influence of varying process parameters on critical depth and surface damage was evaluated through the fluctuations of force during ultrasonic scratching.

#### 3.3.1. Effect of Scratching Speed on Scratching Force

Figure 6 shows the effects of various scratching speeds on normal scratching force and the tangential scratching force of SiC ceramics under an ultrasonic amplitude of 4 μm.

According to Figure 6, the material removal is primarily divided into two stages, namely plastic and brittle removal, and the forces $F_{t}$ and $F_{n}$ increase when the cutting depth increases from small to large. According to Equation (15), the critical depth is increased when the scratching speed decreases, and the plastic removal area is increased accordingly. This is because when the scratching speed is lower, the compression and friction force between the abrasive particles on the tip of diamond pen and workpiece are smaller, and the wear of the indenter is relatively smaller. Under the same scratching conditions, intermittent ultrasonic vibration increased the plastic removal area, and resulted in a more stable scratching force with smaller fluctuations during this stage. Brittle material removal occurs once the normal force exerted by single abrasive particle surpasses its critical load, and the fluctuations of the scratching force during this stage is relatively large and unstable, which indicates that SiC ceramics are mostly removed through plastic removal due to ultrasonic vibrations.

#### 3.3.2. Effects of Ultrasonic Amplitude on the Scratching Force

Figure 7 shows the effects of various amplitudes on the scratching force of SiC ceramics under variable cutting depths. According to Figure 7, forces $F_{t}$ and $F_{n}$ decrease with the increase in the ultrasonic amplitude, and the critical depth increases with the ultrasonic amplitude, then the plastic removal area is increased accordingly. This is because the depth of scratching at the initial stage is lower, and chips are removed by shearing and sliding. When the scratching depth remains below the critical threshold, the force increases steadily with the cutting depth. However, when machining depth transcends critical limits, when cracks appear, the chips will be removed in the form of debris, and the scratching force begins to fluctuate. The impact energy of the indenter increases with the amplitude, which makes it is easy for the workpiece to be removed. As a result, the critical depth of the workpiece is increased, the materials are mainly removed by the plastic formation, and LTUVG minimizes surface cracks, which greatly improve surface quality of SiC ceramics.

#### 3.3.3. Morphological Analysis of the Scratched Surface of Silicon Carbide Ceramic

Figure 8 shows the morphologies of the machined surface of SiC ceramics under normal scratching and ultrasonic scratching with the variable cutting depth. According to Equations (11) and (14), the critical cutting depth under LTUVG conditions is related to grinding parameters, dynamic fracture toughness, and strain rate and it increases with the dynamic fracture toughness and strain rate. According to Figure 8, the scratching width on the machined surface of SiC ceramics increases with the scratching depth of single abrasive particle, whether the process is ultrasonic scratching or normal scratching, and the workpiece is primarily removed by means of brittle removal, so the quality of grooves on the machined surface after normal scratching is poor and the intermittent pits are distributed bilaterally along the groove. However, there are more plastic removal features on the machined surface after ultrasonic scratching, and there are no obvious pits on the scratching surface, so the damage in the machined surface under ultrasonic scratching is less than that under normal scratching conditions. As a result, the surface quality under ultrasonic scratching is significantly improved, and the plastic removal region is expanded, so LTUVG is more suitable for the precision machining of ceramics.

## 4. Conclusions

To investigate the mechanism of low-damage material removal, critical depth was modeled based on the motion trajectory of single abrasive grain and the maximum undeformed cutting thickness of SiC ceramic during LTUVG, and the LTUS test was conducted on SiC to examine the role of ultrasonic vibration in influencing strain rate and dynamic fracture toughness, as well as the different machining parameters impacts governing critical depth and surface quality. Presented below are the primary conclusions:(1)According the kinematic characteristics of LTUVG, LTUV can increase the cutting arc length of single abrasive particle and reduce the maximum unaltered cutting thickness of SiC, and the maximum unaltered cutting thickness decreases with the increase in grinding speed, ultrasonic amplitude, and the cutting arc length and increases with the grinding depth and feed rate.(2)Based on the critical depth of SiC under LTUVG, LTUV greatly improves the strain rate, dynamic fracture toughness, and critical depth; as a result, the plastic removal area increased, and the critical depth increased with the grinding speed, ultrasonic amplitude, and the strain rate of SiC and decreased with the increase in grinding depth and feed rate.(3)In terms of the LTUS tests of SiC, it can be seen that LTUV significantly reduces the scratching force and changes the removal method, and the brittle removal of normal scratching can cause severe damage on both sides of the groove and the large pits on the machined surface; however, ultrasonic scratching expands the plastic removal on the machined surface, there are no obvious pits on the scratched surface, and the quality is significantly improved.(4)Future research should mainly focus on the interactions between multiple abrasive particles and the energy transfer characteristics during LTUVG to explore the material removal mechanism of SiC ceramics, which could provide a theoretical basis for the ultraprecise machining of brittle–hard material.

## Figures and Tables

**Figure 1 micromachines-16-01048-f001:**
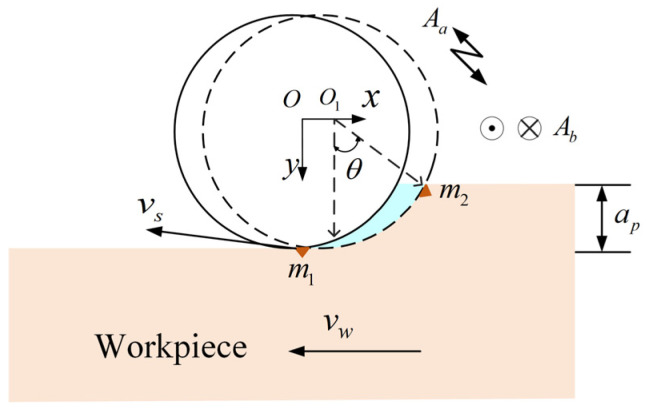
LTUVG system.

**Figure 2 micromachines-16-01048-f002:**
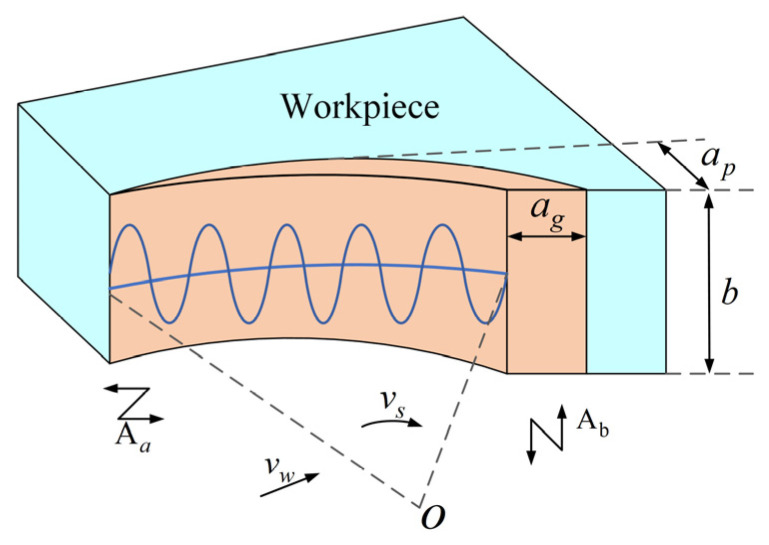
Maximum unaltered cutting thickness schematic.

**Figure 3 micromachines-16-01048-f003:**
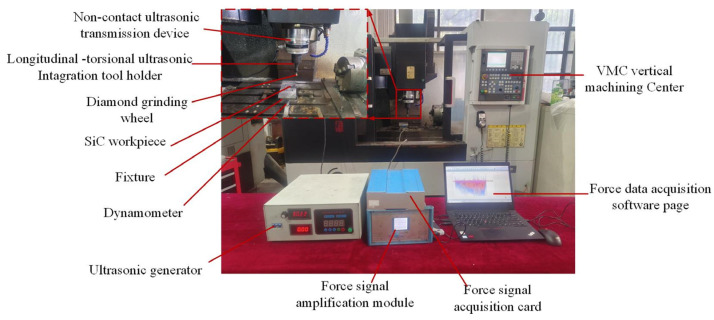
LTUS test platform.

**Figure 4 micromachines-16-01048-f004:**
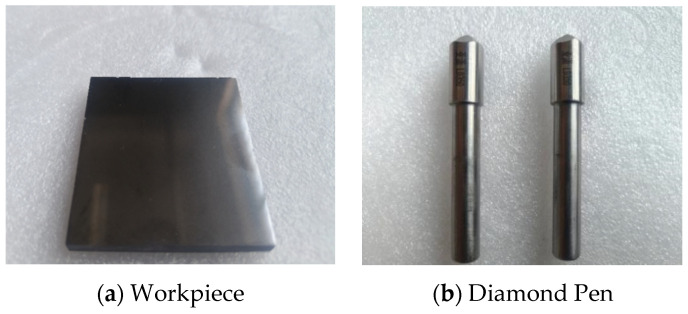
Workpiece and diamond pen.

**Figure 5 micromachines-16-01048-f005:**
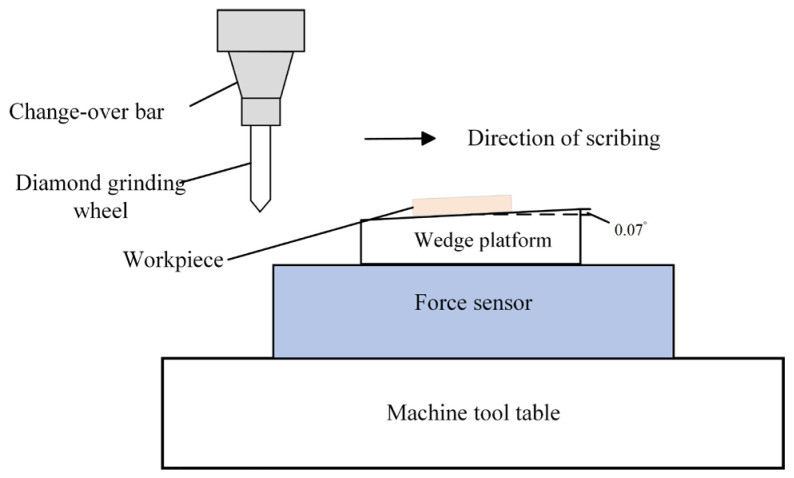
Schematic diagram of ultrasonic scratching.

**Figure 6 micromachines-16-01048-f006:**
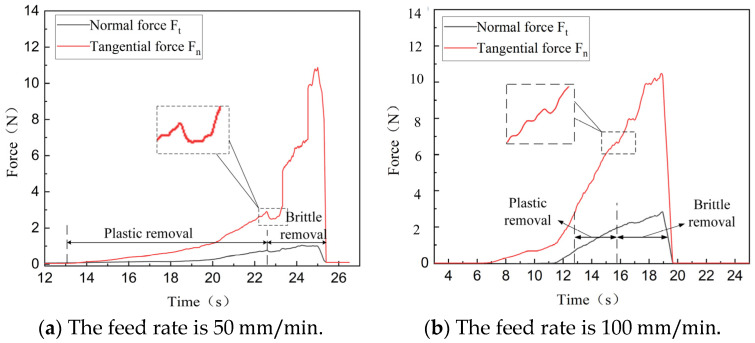
Effect of scratching speed on scratching force.

**Figure 7 micromachines-16-01048-f007:**
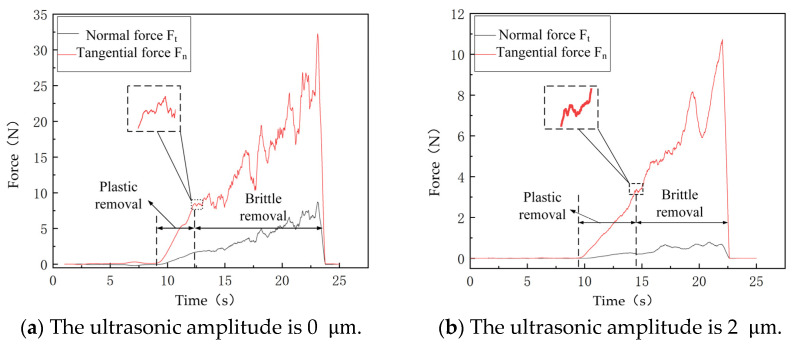
Effect of ultrasonic amplitude on the scratching force.

**Figure 8 micromachines-16-01048-f008:**
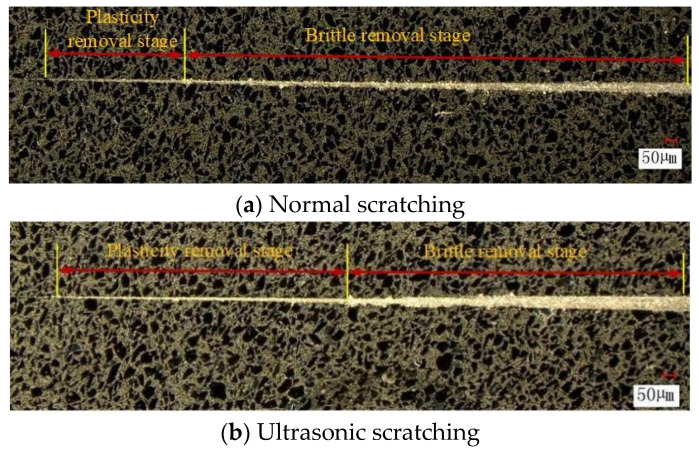
Surface morphology of normal scratching and ultrasonic scratching.

**Table 1 micromachines-16-01048-t001:** Main performance parameters of workpiece.

Material	Density (kg/m^3^)	Hardness (G Pa)	Fracture Toughness (MPa·m^1/2^)	Elastic Modulus (G Pa)	Poisson’s Ratio
SiC	3560	33	5	410	0.14

**Table 2 micromachines-16-01048-t002:** Scratch test parameters.

Group Number	Feed Rate mm/min	Ultrasonic Amplitude (μm)
B1	50	4
B2	100	4
B3	150	4
B4	200	4
B5	100	0
B6	100	2
B7	100	4
B8	100	6

## Data Availability

The original contributions presented in this study are included in the article. Further inquiries can be directed to the corresponding author.

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
