# Peer review of "Study on the Low-Damage Material Removal Mechanism of Silicon Carbide Ceramics Under Longitudinal–Torsional Ultrasonic Grinding Conditions"

_micromachines, 2025, doi:10.3390/mi16091048_

Round 1
Reviewer 1 Report
Comments and Suggestions for Authors
This paper explores the study on the low-damage material removal mechanism of silicon carbide (SiC) ceramics under longitudinal torsional ultrasonic grinding. The work is well-structured, combining theoretical modeling with experimental validation to explore the effects of LTUV on the material removal mechanisms. The research is relevant and contributes valuable insights to the field of precision machining of hard and brittle materials materials. Overall, the article is well organized.
However, some minor issues still need to be improved:
- The annotation text in Figure 5 is too small to read clearly, please correct it and review the entire text to avoid similar situations.
- There are a few grammar errors, Please review and polish the entire text.
- Some letters in the formula are not given the necessary units, such as Va, Vb and Ag. Please add relevant explanations to make reading more smooth.
- The conclusion section should be more concrete and provide a brief description of future research plans.
Reviewer 2 Report
Comments and Suggestions for Authors
The manuscript presents an investigation in scratching SiC material with ultrasonic vibrations. The study is contemporary and well structured but requires a number of revisions.
The force graphs need to be revised so that the scartching area is dominant in the graph. For example, in Fig 6a, zoom in on the area between 12 and 27 sec. Also, the font size needs to be increased to make the labels legible. The graphs when possible should have the same axis range to make comparisons easier. for example figs 6a-c shoould have the same range on x and y axis.
How do the authors justify the large increase between trials B1-3 and the results on trial B4.
What was the sampling rate of the dynamometer.
Was an FFT done on the resulting cutting forces to identify the frequency of the oscilation and the damping effect of the material. Such results should be included on the manuscript.
In results fo trials B5-7, the duration of the test should be identical as the same feedrate was used. On figure 7 the processing times decrease as the ultrasonic amplitude is increased. why is this the case?
The evaluation of the scratches needs to be improved. Comparisons similar to the ones done on the force response should be performed for the geometry of the scratched surfaces.
Round 2
Reviewer 2 Report
Comments and Suggestions for Authors
The authors have addressed all my comments so from my side the manuscript is ok to proceed